# WaSH CQI: Applying continuous quality improvement methods to water service delivery in four districts of rural northern Ghana

Michael B. Fisher[1]*, Leslie Danquah[2], Zakaria Seidu[3], Allison N. Fechter[4], Bansaga Saga[5], Jamie K. Bartram[1], Kaida M. Liang[1], Rohit Ramaswamy[6]*

1 Department of Environmental Sciences and Engineering, The Water Institute at UNC, University of North Carolina at Chapel Hill, Chapel Hill, NC, United States of America, 2 School of Geosciences, University of Energy and Natural Resources, Sunyani, Ghana, 3 West African Centre for Cell Biology of Infectious Pathogens, University of Ghana, Legon, Ghana, 4 The Water Project, Concord, NH, United States of America, 5 Solidarites International, Clichy, France, 6 Public Health Leadership Program, Gillings School of Global Public Health, University of North Carolina, Chapel Hill, NC, United States of America

* mbfisher@gmail.com (MBF); ramaswam@email.unc.edu (RR)

**Data Availability Statement:** All relevant data are within the manuscript and its Supporting Information files.

## Abstract

Continuous, safely managed water is critical to health and development, but rural service delivery faces complex challenges in low- and middle-income countries (LMICs). We report the first application of continuous quality improvement (CQI) methods to improve the microbial quality of household water for consumption (HWC) and the functionality of water sources in four rural districts of northern Ghana. We further report on the impacts of interventions developed through these methods. A local CQI team was formed and trained in CQI methods. Baseline data were collected and analyzed to identify determinants of service delivery problems and microbial safety. The CQI team randomized communities, developed an improvement package, iteratively piloted it in intervention communities, and used uptake survey data to refine the package. The final improvement package comprised safe water storage containers, refresher training for community WaSH committees and replacement of missing maintenance tools. This package significantly reduced contamination of HWC (p<0.01), and significant reduction in contamination persisted two years after implementation. Repair times in both intervention and control arms decreased relative to baseline (p<0.05), but differences between intervention and control arms were not significant at endline. Further work is needed to build on the gains in household water quality observed in this work, sustain and scale these improvements, and explore applications of CQI to other aspects of water supply and sanitation.

## Introduction

Continuous access to adequate quantities of safe water is critical to human health and development [1]. A substantive burden of disease is associated with inadequate water services in low-

**Funding:** This work was supported in part by a grant from The Conrad N. Hilton Foundation https://www.hiltonfoundation.org/. This grant was awarded to JKB and also supported KL, MBF, ANF, and RR. This work was also supported in part by a grant from World Vision https://www.worldvision.org. This grant was awarded to JKB and also supported KL, MBF, and ANF. In addition, this work was supported in part by a grant from the National Institute of Environmental Health Sciences (T32ES007018), which supported MBF in part: https://www.niehs.nih.gov/index.cfm. World Vision played the following role in the design of the study: World Vision personnel (including Bansaga Saga, who worked for WV at the tiime the study was designed) reviewed a summary of the study design, but did not seek to modify the study design. World Vision personnel also played a role in data collection, under the guidance of the authors. None of the funders played a role in data analysis or the decision to publish. Bansaga Saga, who previously worked for World Vision, but no longer worked for any funder at the time of manuscript preparation and submission, also played a role in the preparation of the manuscript.

**Competing interests:** The authors have declared that no competing interests exist.

income country (LIC) and middle-income country (MIC) settings [2]. Considerable progress has been made in expanding access to safe water in recent decades [3]. However, disruptions in service [4], as well as microbial contamination of water during transport and storage [5–7] deny safe water to families in rural low- and middle-income country (LMIC) settings [8]. In 2015, the Sustainable Development Goals (SDGs) were adopted by UN member states. SDG Six on drinking water and sanitation calls for universal coverage of drinking water and sanitation and improvements in levels of service. The continuity and safe management of drinking water services, both at the source and household levels, are important to achieving this goal and are incorporated into the language of the targets. While countries have begun working to achieve these targets, many challenges prevent the continuous availability of safely managed water at the household level in rural low-and middle-income-country (LMIC) settings. In Ghana (a lower- middle-income country), as in much of rural sub-Saharan Africa, use of communal improved water sources is widespread. As of 2015, 84% of households in rural Ghana used an improved primary water source [3], while only 7% used a primary water source that was on-premises [9], with the balance largely relying on communal sources. While many of these sources provide basic access [9], service discontinuity and microbial contamination create persistent challenges to water quality in the home [10, 11].

Improving safely managed water services presents complex challenges [10] because service continuity and water safety depend on context-specific technical, social, geographic, and behavioral factors [12]. To sustain improvements, evidence-based solutions must be adapted to local needs and conditions. Continuous Quality Improvement (CQI) methods such as Lean [13], Six Sigma [14], and the Model for Improvement [15] were developed in manufacturing [17] and are now widely applied to health care in both high-income [16, 17] and in low- and middle-income settings [18–21]. These methods have successfully engaging local teams to develop context-appropriate solutions to improve system performance across disciplines. However, CQI has not been systematically applied to complex water supply and sanitation challenges in low-income settings (such as much of SSA) or middle-income settings such as Ghana.

While industrial and health-care improvements are often implemented in centralized and controlled settings, water supply and sanitation programs in low- and lower-middle income settings are often implemented in diverse and decentralized community settings; "one-size-fits-all" solutions are rare, and there is often a gap between evidence and current practice. CQI methods are well suited to addressing deficiencies in such programs. They engage community members to combine evidence and monitoring data with local knowledge to systematically identify, adapt, and implement improvement packages to a given context. However, there is limited evidence on how best to apply CQI to community-based health programs [22], and the application of CQI to rural community water supply and sanitation challenges in LMICs has not been described previously.

This work addressed two objectives:

1. The application of community-based CQI to reducing microbial contamination of stored water in rural households across four districts in Ghana; and

2. The application of these CQI methods to improving the functionality of handpumps attached to boreholes in this setting.

In this paper, we assess the effectiveness of CQI methods in these applications (using process indicators such as measures of uptake), the performance of the resulting interventions (using targeted outcome indicators), and lessons learned for future potential applications of CQI methods to similar challenges in water supply and sanitation across low- and lower-

middle income settings. Because CQI is inherently driven by local needs, knowledge, and context, a CQI approach may be suitable across a range of settings and challenges to produce context-specific process improvements, even where the specific improvement packages identified through this approach may differ depending on the characteristics of each problem, population, and context.

## Methods

### Context

The CQI approach was piloted in 216 communities in four districts of the Northern Region of Ghana (Savelugu-Nanton, Tolon, Gushiegu, and Karaga districts) by World Vision Ghana (WVG, an international NGO active in this region) in collaboration with The Water Institute at the University of North Carolina at Chapel Hill (UNC WI). These communities were randomly sampled from 296 communities in which WVG had previously implemented water, sanitation, and hygiene (WaSH) programs. These programs included construction of communal water sources (primarily boreholes with handpumps), training of water and sanitation management committees (WSMTs, or WaSH committees), and, in some cases, hygiene and/or sanitation activities.

### Study design

The study used a 2-parallel-group randomized design with 1:1 allocation comparing communities that implemented interventions for improving water quality and access (developed using a CQI process) and those that did not. Analyses included comparison of outcomes between groups at various time points during the implementation, as well as pre-post analysis of outcomes within groups (Additional details in S1 File).

### Ethical approval

Ethical approval was obtained from the Institutional Review Board (IRB) at UNC (Study# 14–0386). In-country ethical approval was obtained from the IRB at the Navrongo Health Research Center (Navrongo, Ghana). Written informed consent was obtained from all participants in household surveys, and identifiable personal data were kept confidential according to standard protocols for human subjects' research.

### CQI process

The CQI process was based on the Six Sigma improvement methodology [14]. This methodology was adapted to emphasize the importance of iterative implementation and sustainability to rural community-based water supply programs. Specifically, the Improve phase was separated into two parts, *Identify* and *Implement* and the Control step was redefined as a *Sustain* step (Table 1, S2 File). An implementation guide was developed for this work [5].

**CQI team formation and project charter creation (DEFINE).** The CQI team comprised WVG staff; UNC WI researchers participated as coaches and facilitators. The team received 5 days of training in CQI methods; at the end of which the team decided to focus improvement efforts on household microbial water quality and handpump functionality. A project charter was created (S3 File) and the team conducted process mapping of current water supply implementation and maintenance practices.

**Survey instruments, sampling and baseline data collection (MEASURE).** Survey instruments (S4 File) were developed and validated according to WaSH monitoring and evaluation best-practices [23]. These included a community-level survey (administered to WSMTs, if

**Table 1. Summary of the adapted WaSH CQI process.**

| Step | Purpose | Activities | Setting | Outputs |
|---|---|---|---|---|
| **DEFINE** | • Form CQI team<br>• Select improvement area of focus | • Assemble Team<br>• Train team in CQI methods<br>• Develop Charter | Office | • Project Charter<br>• Improvement goals |
| **MEASURE** | • Identify key process variables<br>• Create and validate measurement tools<br>• Collect data | • Collect household and water point data | Office and field | • Data collection plan<br>• Validated survey tools<br>• Survey data |
| **ANALYZE** | • Identify correlates of poor performance | • Analyze data using statistical tools. | Office | • Identified root causes of poor performance |
| **IDENTIFY** | • Select and assemble potential improvement solutions | • Review evidence base<br>• Adapt for local conditions<br>• Develop local solutions<br>• Create improvement package | Office | • Prototype improvement package |
| **IMPLEMENT** | • Iteratively implement and refine improvement package | • Implement prototype improvement package in pilot communities<br>• Collect uptake data and iteratively refine improvement package<br>• Assess whether improvement has occurred | Field | • Final improvement package<br>• Midline and endline monitoring data |
| **SUSTAIN** | • Standardize, sustain, and scale improvements | • Develop standard operating procedures<br>• Create training tools<br>• Develop a scale-up plan | Field and Office | • Scale-up plan |

present, or to community leaders if no WSMT was present). A water source survey was conducted at each communal drinking water source in the community (S1, S4 and S5 Files). A household survey captured information on water, sanitation, and hygiene in each household. Surveys were piloted by the CQI team in 5 test communities outside the study area, refined based on pilot experiences, and administered using the Akvo FLOW V 1.6 mobile survey tool on mobile phones running the Android operating system. Use of mobile survey tools in WaSH has been reviewed previously [24, 25].

A sample of stored household water for consumption (HWC) was tested to determine the most probable number (MPN) of *E. coli* (an indicator of microbial contamination) per 100 mL as part of each household survey. Samples were collected in sterile 100-mL Whirl-pak® Thio-bags (Nasco, Ft Atkinson, WI) and enumerated using compartment bag tests (CBT, Aquagenx, LLC, Chapel Hill, NC) [26] with 24-h ambient temperature incubation (ambient temperatures ranged from 30–35 C during the study period).

A sample of 230 communities was selected to enable detection (with 80% power at the 95% confidence level) of: a) a 10% difference in the proportion of household stored water samples having detectable microbial contamination, and b) a 10% difference in handpump functionality between study arms (S6.4 a & b Table in S6 File). The operational definition of functionality used was *a handpump that enabled a 20-L container to be filled within 10 minutes* (minimum threshold for any water availability, as compared to national performance standard of 13.5 L/min [27]; note that the implication of this threshold is not that a yield of 2 L/min is necessarily sufficient to meet community needs, but rather that a binary distinction between sources providing some water vs those providing little or no water is useful [since flow rate is also captured as a separate continuous variable], and 10 minutes represented an indicative upper limit on the amount of time that users and/or enumerators could be anticipated to spend attempting to measure flow). Calculations relied on WVG estimates of typical numbers of water sources and households in communities within the four selected districts. Estimates of baseline water source functionality and household stored water quality were based on a review of published studies from Ghana and other contexts (S6.4 Table in S6 File) [6, 11]. Sampled communities were randomly assigned to intervention or control arms.

Baseline data were collected in 216 of the 230 sampled communities at the outset of the CQI project (S6.1–2, S6.9–11 Table in S6 File). Fourteen communities could not be reached due to poor road conditions or flooding. Water sources were surveyed and WSMTs were interviewed in all visited communities; household surveys were conducted in a random subset of 50% of communities, with 6 households per community selected at random (S1 and S4 Files). This proportion (50% of communities rather than 100%) was chosen based on the estimated sample size required to detect a 10% change in the proportion of households with detectable contamination in stored water samples, and based on the greater time investment required for household surveys. Households with a consenting adult respondent and one or more children under five years old were included: Female heads of household were preferred respondents based on their typically greater involvement in and knowledge of water collection, water management, and childcare practices in the household relative to other household members; if not available, another adult in the household was interviewed. Median survey completion times were approximately 10, 20, and 25 minutes for waterpoint, WSMT, and household surveys, respectively.

**Baseline data review and root cause analysis (ANALYZE).**   Baseline data were analyzed to determine the status of WaSH services in sampled communities (S6 File). The CQI team reviewed preliminary results and verified that household stored water quality and handpump functionality remained improvement priorities. Regression analysis was performed to study associations of targeted outcomes with potential determinants captured in water source and household surveys at baseline. Specifically, multivariable linear regressions were used. In addition, chi$^2$ tests were used to compare the proportions of intervention and control households with microbial contamination at baseline. Stored household water quality was associated with source type, water storage conditions (e.g. storage container opening [wide/narrow], etc.), and household hygiene and sanitation practices; Water source functionality was associated with "non-modifiable" characteristics such as district, week (as a proxy for rainfall), and the number of other sources in the community, as well as "modifiable" management factors such as savings in excess of USD $100 (S6 File: S6.12 Table) (S6 File: S6.13 Table). Most households obtained water from a communal source. Observations indicated this water was typically transported on the head, poured into large storage containers in a central courtyard, and scooped out when needed (S6 File: S6.1 Table, S2 Fig).

**Improvement package development (IDENTIFY).**   The CQI team used structured decision-making tools and participatory methods (Brainstorming, multi-voting, Pugh Matrix [S7 File], focus groups) to develop an improvement package comprising interventions to improve household water quality and water point functionality. These interventions targeted modifiable causes of water quality and functionality issues identified in ANALYZE, as well as other factors identified by the CQI team as potentially important for proposed improvements, despite no association with target outcomes at baseline (e.g. availability of tools). Elements of the final improvement package are shown in Fig 1.

**Implementation, iterative refinement and performance monitoring (IMPLEMENTS).** Safe Water Storage Containers (SWSCs) including taps and tightly fitting lids were manufactured locally and provided to six randomly selected households in each of three randomly selected communities (along with training in their proper use). An initial WSMT refresher training program was developed (based on existing initial training curricula), and delivered to WSMTs in the same communities, along with any required replacement tools needed for water source repair (any essential tools that WSMTs lacked or had broken). Household- and community-level uptake surveys (S4 File) were then conducted in the three test communities, as well as three randomly selected control communities, to assess uptake and performance of the initial improvement package. Data were analyzed as described above. The improvement

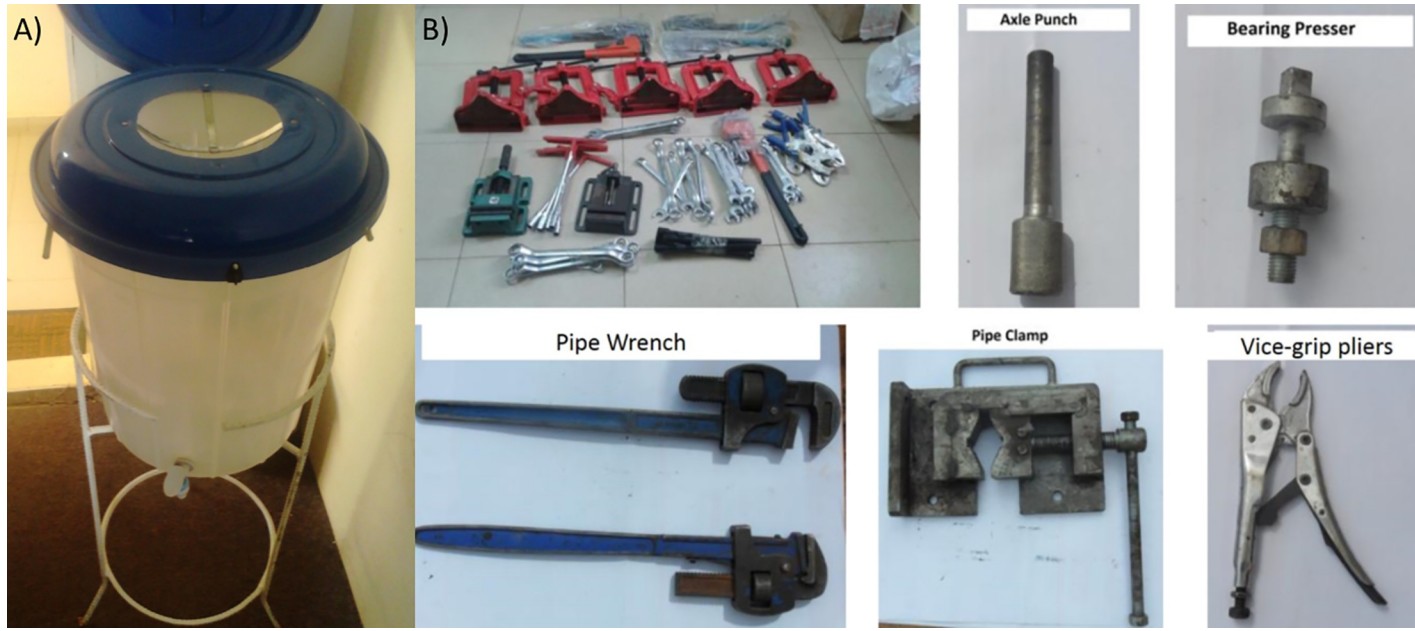

**Fig 1.** Improvement Package Components: A) Safe water storage container; B) Tools for water system repair.

package was refined based on these findings (V 1.1), implemented in three additional test communities, and the process was iterated in successive implementation rounds (Table 3) until a final improvement package was identified (S8 File) and scaled to remaining intervention communities. The final package included refined SWSCs and user instructions, "clustered" WSMT refresher training (3–10 WSMTs per training), and replacement tool distribution. Many replacement tools were unavailable from local vendors (at required quantity and quality), and

**Table 2. Multivariable regression of *E. coli* presence in household stored water quality at baseline, controlling for district and week of sample collection.**

| Variable | Odds Ratio | P>z | [95% Conf. | Interval] |
|---|---|---|---|---|
| **Source Type**** | | | | |
| **Borehole with handpump** | 1 | | | |
| **Piped water into dwelling** | 0.0786 | *0.077* | 0.00470 | 1.313 |
| **Public tap/standpipe** | 0.999 | *0.999* | 0.386 | 2.588 |
| **Rainwater collection** | 0.434 | *0.017** | 0.219 | 0.863 |
| **Surface water** | 1.283 | *0.564* | 0.551 | 2.987 |
| **Unprotected dug well** | 1.214 | *0.718* | 0.423 | 3.483 |
| **Hh storage container has lid** | 0.457 | *0.018** | 0.238 | 0.875 |
| **Hh storage container is narrow** | 0.972 | *0.076* | 0.941 | 1.003 |
| **Hh has a latrine** | 0.445 | *0.045** | 0.201 | 0.982 |
| **Handwashing with soap observed** | 0.936 | *0.815* | 0.538 | 1.628 |
| **Handwashing observed with a rubbing motion** | 0.303 | *0.003** | 0.138 | 0.668 |
| **Where are child's feces emptied** | | | | |
| **At refuse dump** | 1 | | | |
| **Dig and bury** | 0.0714 | *0.094* | 0.00325 | 1.566 |
| **[Bush, field, no sanitation facilities]** | 2.586 | *0.399* | 0.285 | 23.467 |

*Significant at 95% Confidence Level (CL)

** Significant at 99% CL

**Table 3. Implementation rounds.**

| Round | District | Intervention communities* | Control communities* | Changes made | Rationale (from uptake surveys and field observations) |
|---|---|---|---|---|---|
| 1 | Savelugu | 3 | 3 | Modify SWSC base; Clustering of WSMT refresher training activities | High rate of breakage; Increase training efficiency |
| 2 | Savelugu | 3 | 3 | Modify SWSC lid to allow pouring water in but prevent dipping/scooping water out | Users bypassing tap (dipping water) |
| 3 | Savelugu | 3 | 3 | Further modify SWSC lid | Continued dipping observed |
| 4 | Tolon | 9 | 11 | Replacement of low-quality locally manufactured tools | Selected tools observed to perform poorly, break frequently |
| 5 | All | 109 | 107 | No further changes | |

SWSC = Safe Water Storage Container; WSMT = Water and Sanitation Management Team

*Note that in each implementation round, the specified numbers of intervention and control communities are new communities added in that round (and different from those in previous rounds)

delays in international procurement led to delivery after midline data collection for many communities. The changes made in each iteration are shown in Table 3.

**Developing standard operating procedures (SUSTAIN).** At the end of the refinements, standard operating procedures (SOPs) for the incorporation of these improvements into WaSH programs were developed. Two rounds of post-implementation monitoring were conducted with the same communities and households as at baseline, to assess impacts of the improvement package. Some loss to follow-up occurred in each subsequent round of monitoring, as 11% and 18% households visited at baseline were unavailable at midline and endline, respectively (Table 4). Data analyses were conducted at endline in a similar manner to those described at baseline. A slightly larger proportion of selected communities were reachable at endline (216) compared to baseline and midline as a result of changes in road conditions and other logistical factors across sampling periods. Summary statistics on outcomes of interest stratified by treatment condition and monitoring round were calculated. Summary statistics at baseline were compared across treatment conditions to assess randomization. Univariable regressions of outcomes as a function of treatment assignment (intention-to-treat) and treatment delivered (per-protocol) were conducted. Since outcomes of interest could be affected by factors other than the interventions, multivariable regressions were also performed that included process variables associated with the intervention package as independent variables and controlled for geography, week of assessment (as a proxy for rainfall) and other covariates.

# Results

## Baseline results

Baseline results (S6 File: S6.1–2 Table) show poor household stored water quality, as well as a substantive proportion of water sources with detectable *E. coli*. Approximately two thirds of

**Table 4. Performance monitoring rounds: Numbers of communities, water sources, and households captured in each round (inclusive of both intervention and control arms, and of pilot communities captured in Table 2).**

| Round: | Baseline | Midline | Endline |
|---|---|---|---|
| Communities | 212 | 205 | 216 |
| Water sources | 926 | 924 | 983 |
| Households | 527 | 471 | 431 |
| Completion Date | November 1, 2014 | November 1, 2015 | May 1, 2017 |
| Loss to follow-up (HH) | – | 10.6% | 18.2% |

**Table 5. Uptake statistics by implementation round (Household level).**

| Variable | Round 1 | Round 2 | Round 3 | Round 4 |
|---|---|---|---|---|
| **Dates** | Aug 2014 | Sept 2014 | Feb 2015 | Mar 2015 |
| | % (n) | % (n) | % (n) | % (n) |
| **Report receiving Safe Water Storage Container** | 100% (15) | 100% (18) | 100% (19) | 98% (50) |
| **SWSC has water in it** | 73% (15) | 72% (18) | 100% (19) | 92% (50) |
| **SWSC broken or cracked** | 7% (14) | 44% (18) | 0% (19) | 4% (50) |

SWSC = Safe Water Storage Container

boreholes with handpumps were functional on the day of the visit. Most communities had a WSMT. The average time since WSMTs received training was over 5 years. Characteristics of intervention and control communities and households were largely similar at baseline (S6 File: S6.1–3 Tables). However, more control than intervention households reported water continuously available at baseline (80% vs 70%, p = 0.009) and more intervention than control households reported treating their water at baseline (25% vs 17%, p = 0.03). No significant differences in water source characteristics were observed across treatment arms.

**Uptake results during implementation.** Table 5 shows uptake survey results by implementation round. Uptake data and enumerator observations indicated that initial prototype SWSCs were prone to tipping and breakage; many did not show signs of recent use; and many users continued to remove water by dipping or scooping (high contamination risk), despite the presence of a tap. SWSCs were redesigned to enhance stability and ease of use (support redesigned) and prevent dipping while still enabling users to fill from containers carried on the head/shoulder (opening redesigned). Following these iterative refinements, the proportion of containers with water in them increased from <75% in rounds 1–2 to >90% in rounds 3 and 4). Enumerators reported in subsequent rounds that later SWSC variants were increasingly stored outside, where activities related to water consumption traditionally take place in northern Ghana. Container breakage rates also decreased.

**Post-implementation uptake results.** Tables 6 and 7 and S6.5–6 show uptake data from post-implementation monitoring. Data were collected 6–12 months after implementation for midline and two years after implementation for endline. Uptake of improvement package elements was high: 105 out of the 109 invited WSMTs participated in refresher trainings between baseline and midline, and tools were delivered to all intervention communities that were missing tools between baseline and endline (due to procurement delays). At endline, 79% of intervention communities reported having all tools needed to maintain water sources, vs 43% in control communities (*p<0.01*, S6 File: S6.5 Table). Safe water storage containers were delivered to all intervention households. At midline, SWSCs were observed in 86% of intervention households (measured as a proxy for implementation fidelity, sustained uptake, and container

**Table 6. Proportion of households with safe water storage container by treatment group.**

| Time point | Baseline | Midline | Endline |
|---|---|---|---|
| **Intervention** | 0% (208) | 86% (242) | 57% (n = 197) |
| **Control** | 0% (270) | 10% (225) | 17% (n = 234) |
| **Pearson Chi$^2$ (p)** | N/A | 270.7 (*0.000***) | 74.1 (*0.000***) |

*Results significant at 95% confidence level

**Results significant at 99% confidence level

**Table 7. Proportion of household water samples in the high-risk category by treatment group (intention-to-treat and per-protocol).**

| a) Intention-to-treat | | | |
|---|---|---|---|
| **Treatment** | Baseline | Midline | Endline |
| **Intervention (assigned)** | 53% (214) | **35% (237)** | 34% (n = 194) |
| **Control (assigned)** | 53% (263) | **50% (213)** | 42% (n = 232) |
| **Pearson Chi² (p)** | 0.0001 (*0.992*) | 10.6 (*0.001*\*\*) | 2.7 (*0.099*) |
| b) As treated | | | |
| | Baseline | Midline | Endline |
| **Safe storage (observed)** | N/A | **35% (222)** | **30% (n = 95)** |
| **Other Storage (observed)** | 52% (512) | **50% (232)** | **43% (n = 331)** |
| **Pearson Chi² (p)** | N/A | 10.3 (*0.001*\*\*) | 7.1 (*0.008*\*\*) |
| c) As treated, improved source | | | |
| | Baseline | Midline | Endline |
| **Safe storage, improved (observed)** | N/A | **35% (221)** | 32% (n = 81) |
| **Other (observed)** | 53% | **50% (232)** | 40% (n = 345) |
| **Pearson Chi² (p)** | N/A | 10.1 (*0.002*\*\*) | 1.6 (*0.205*) |

\*Results significant at 95% confidence level

\*\*Results significant at 99% confidence level

survival). This figure decreased to 57% by endline (Table 6). Meeting frequency of WaSH committees did not change significantly across time points or treatment arms (S6 File: S6.6 Table).

**Outcome results.** Midline and endline monitoring showed significant improvements ($p<0.10$) in HWC quality among intervention communities vs control communities (intention-to-treat level, Table 7A), and significant improvements ($p<0.01$) among households with SWSCs vs households without safe storage containers (as treated, Table 7B), particularly for households using an improved water source (Table 7C).

S6 File: S6.7 Table shows the effect of source water on HWC quality. At both midline and endline, households with SWSCs that used an improved water source (e.g. borehole with handpump or piped water) as their primary source of HWC were still less likely to be in the high-risk category (*E. coli* MPN > = 100 CFU/100 mL).

When other factors are controlled for, a significant increase in functionality across both groups between endline and baseline was observed (Table 8), and a significant association between functionality and access to tools was also observed (S6 File: S6.8 Table). By contrast, a simple pre-post test did not show significant differences in handpump functionality across timepoints (Table 9).

## Discussion

### Relevance of CQI to WaSH challenges

The United Nations launched the Sustainable Development Goals (SDGs) in September 2015, to replace the Millennium Development Goals. SDG 6 on drinking water and sanitation calls for universal coverage of drinking water and sanitation and improvements in levels of service. The continuity and safe management of drinking water services, both at the source and household levels, are important to achieving this goal and are incorporated into the language of the targets. While countries have begun working to achieve these targets, many challenges prevent the continuous availability of safely managed water at the household level in rural low-and middle-income-country (LMIC) settings such as northern Ghana. Continuous Quality

**Table 8. Multivariable logistic regression of water source functionality in intervention vs control communities controlling for district, week, and number of users per waterpoint (n = 1586 across 3 monitoring rounds).**

| Variable | Odds Ratio | Std. error | Z | P>z | 95% CI |
|---|---|---|---|---|---|
| Community Type (Intervention vs. Control) | 1.177 | 0.149 | 1.28 | 0.199 | 0.918–1.509 |
| Monitoring Round | | | | | |
| 2 vs 1 | 1.041 | 0.279 | 0.15 | 0.882 | 0.615–1.759 |
| 3 vs 1 | 3.569 | 2.632 | 1.73 | 0.084* | 0.841–15.15 |
| Source Type | | | | | |
| Borehole with handpump | 1 (Reference) | - | - | - | - |
| Mechanized borehole | .1456 | 0.104 | -2.79 | 0.007** | 0.036–0.590 |
| Piped water into dwelling | 0.4430 | 0.153 | -2.36 | 0.018* | 0.225–0.870 |
| Public tap/standpipe | 0.5462 | 0.089 | -3.72 | 0.000** | 0.397–0.751 |
| Water points per community (+1) | 0.8800 | 0.017 | -6.81 | 0.000** | 0.848–0.913 |
| Seasonality (Seasonal Unavailability) | 0.1239 | 0.029 | -8.96 | 0.000** | 0.0784–0.196 |
| Model Chi² statistic | 328.33 | | | | |
| Model Prob > Chi² | 0.0000** | | | | |

*Results significant at 90% confidence level

**Results significant at 95% confidence level

Improvement (CQI) methods are well established in manufacturing, health care, and other sectors, but have not been previously applied to water and sanitation in rural LMIC settings.

**Overall findings.** By engaging stakeholders in systematic problem solving using local data, CQI enables the identification and implementation of solutions that fit local contexts. Improvement packages combining prior knowledge and evidence with local innovations adapted through field testing are better able to be adopted and sustained than those developed based on prior knowledge alone. In low- and lower-middle income countries, CQI has primarily been used to improve outcomes in health care facilities. To the best of our knowledge, this work is the first attempt to implement CQI in a rural lower- or middle-income community setting, and the first adaptation of these methods to WaSH in such settings. This work demonstrates that CQI can be used to develop solutions to such challenges in northern Ghana, and potentially other settings as well.

The final version of the SWSC was significantly modified from initial prototypes: uptake data and enumerator observations supported iterative testing and refinement, leading to lower breakage rates, greater uptake, and less dipping/scooping (which contribute to contamination). Two years after intervention implementation, half of intervention households were using SWSCs, and these households were less likely to have highly contaminated household stored water than control households (Table 7A). The use of SWSCs incrementally improved water safety: 30% of households using SWSCs consumed water in the high-risk category,

**Table 9. Proportion of boreholes with handpumps functioning on the day of the visit by treatment group.**

| Time point | Baseline | Midline | Endline |
|---|---|---|---|
| Intervention | 67% (n = 446) | 67% (n = 414) | 55% (n = 435) |
| Control | 62% (n = 473) | 61% (n = 465) | 52% (n = 493) |
| P (Chi²) | 0.161 | 0.044 | 0.289 |

*Results significant at 90% confidence level

**Results significant at 95% confidence level

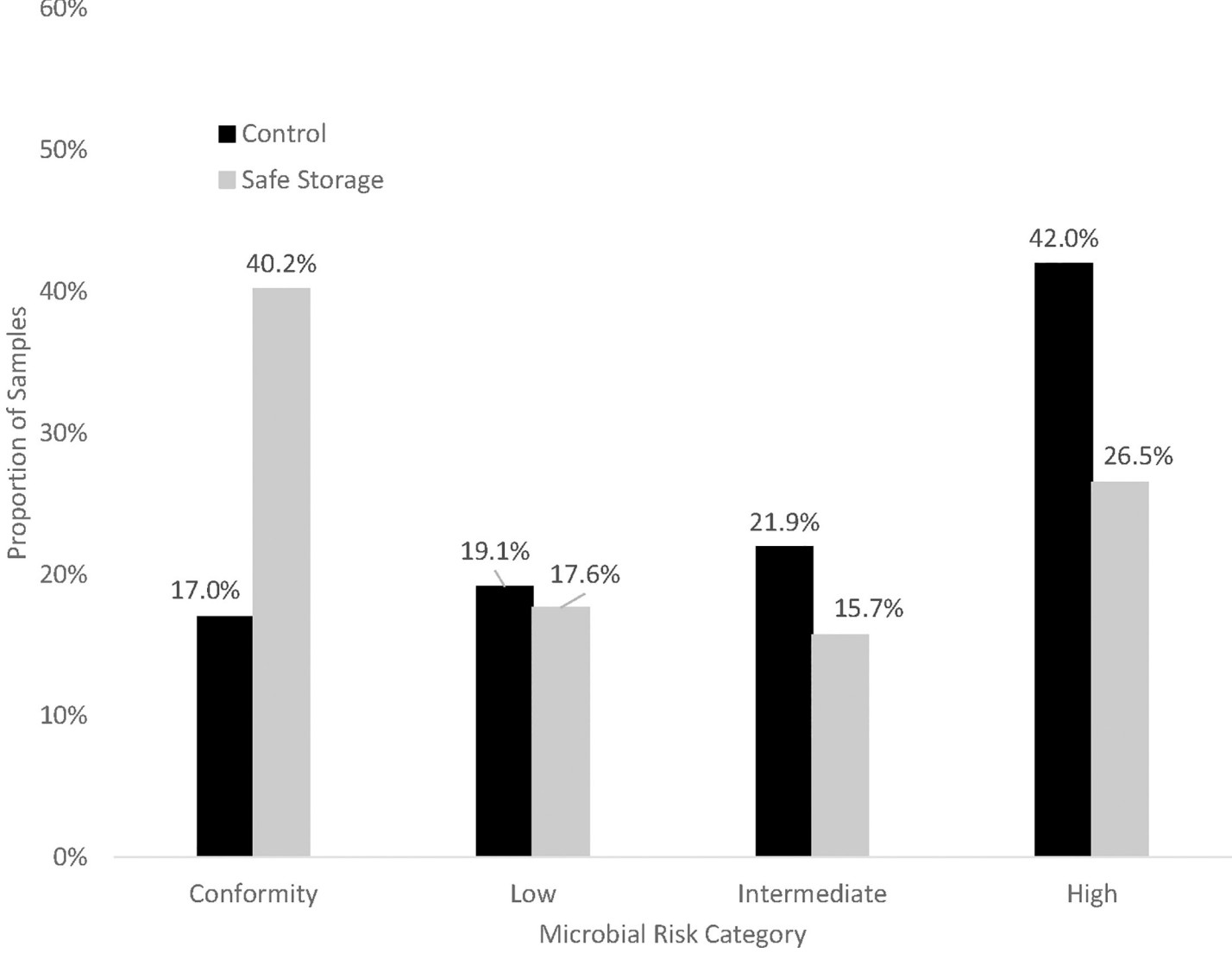

**Fig 2. Microbial risk category of household water for consumption vs treatment (as-treated) at endline.**

compared to 43% of households without SWSCs (Table 7B, Fig 2, p<0.01). CQI produced locally acceptable and effective SWSCs that performed better than containers available prior to this structured, iterative implementation approach. Further efforts may be needed to ensure that safely stored water remains free from fecal contamination in all households, and improvements targeting water treatment in this context may be of interest. Furthermore, the reduction in the proportion of intervention households with SWSCs over time may be due to loss and/or breakage, and further work may explore options to further enhance durability and desirability of SWSCs.

The functionality results are less clear. The CQI team implemented refresher training and distribution of missing tools in intervention communities, and this package was refined through iterative implementation. WSMTs were initially trained individually or in clusters of 2–3 communities at a location within one of the communities. However, CQI team members reported that WSMTs from communities with high-ranking chiefs were reluctant to travel to

nearby communities with lower ranking chiefs for training. The CQI team found that hosting training sessions at neutral locations (e.g. schools or WVG facilities) eliminated chieftaincy concerns; "clustering" 5–10 WSMTs at each training session became possible in such locations, saving time and resources. Furthermore, iterative implementation enabled the CQI team to identify which replacement tools could be sourced locally, and which local tools (e.g. pipe clamp, rod-lifter) were of inadequate quality and prone to breakage and/or malfunction, and therefore needed to be sourced internationally.

As the results in S6.8 Table in S6 File indicate, access to tools was associated with functionality once environmental and community factors are included. It is therefore possible that the solution suggested by the CQI team was appropriate, and that failure to detect a difference in treatment arms at the intention-to-treat level was related to limitations in implementation fidelity, not the intervention delivered.

Furthermore, the finding that overall functionality improved between baseline and endline after controlling for relevant covariates, but was not significantly different between intervention and control communities (Table 8) suggests that the benefits of WSMT refresher training and/or replacement tools may have spilled over to control communities. Enumerators reported anecdotally hearing many instances of WSMTs providing support to or receiving assistance from nearby communities.

**Limitations and implementation challenges.**    Several limitations and implementation challenges characterized this work.

a. *Challenges in implementing CQI*: Engagement of committed organizational leadership within the implementing organization was essential to implementing CQI; obstacles included an organizational structure in which many activities required multiple approvals —these could result in delays unless actively "pushed" forward by internal champions of adequate rank. Furthermore, during peak periods of operational activity, organizational capacity to implement CQI activities was diminished. While CQI is designed to integrate into the everyday problem-solving approach of an organization, improvement was instead viewed as a separate, special project by some participants; organizational leadership worked to modify this perception, but with limited progress. As a result, the team's ability to regularly engage staff members in improvement activities was sometimes constrained. Furthermore, existing monitoring capacity within the implementing organization was augmented through capacity building and recruitment as part of the CQI work; sustaining and scaling this capacity represents an independent challenge to implementing rigorous CQI activities.

b. *Local Tools Unsuitable*: As noted above, certain locally manufactured tools were of inadequate quality and prone to breakage/malfunction—these needed to be replaced with imported items that performed better in the field, resulting in delays and increased costs.

c. *Logistical constraints*: One major challenge of the study was the logistic complexity and cost of longitudinal data collection and iterative improvement implementation among the selected number of geographically dispersed communities. Implementation of CQI monitoring was relatively involved, with travel times of 1–2 hours to reach many communities, and survey collection times were as described in methods. The cost of data collection was on the order of USD$100 per community, with transportation time and fuel representing a substantive proportion of this cost. Given the cost of reaching communities, the incremental cost of each survey question was low; furthermore, detailed data collection on process and outcome variables was of particular interest given the lack of high-quality evidence on WaSH CQI in rural LMIC settings. However, future WaSH CQI efforts in rural settings may target smaller numbers of communities in one or more clusters in each round, and

may streamline data collection tools where appropriate, to accelerate iterative implementation. The use of remote sensors, telephone or SMS uptake surveys, and/or other rapid data collection methods may be useful for obtaining higher frequency data without overburdening communities and implementers, while reducing costs.

d. *Documentation and recall challenges*: As noted above, several intervention households did not have SWSCs at follow-up, while 2% of control households had containers meeting the definition of SWSCs at midline (survey photos suggest that most of these were not distributed as part of the current study). Furthermore, while 95% of intervention WSMTs and 0% of control WSMTs reported participating in refresher trainings, some control community WSMTs reported receiving recent trainings, while many intervention community WSMTs reported that they had not. This discrepancy was most likely due to recall errors, but it is also possible that some unintentional design contamination occurred. It is likewise possible (though unlikely) that some intervention communities may have knowingly under-reported activities in hopes of receiving additional training and/or support.

e. *Spillover*: As noted above, intervention-community WSMTs who received refresher training and tools may have contributed to the maintenance of water sources in nearby control communities ("spillover"). The potential spillover of training and tools (e.g. through informal "mutual support") is potentially advantageous and adaptive with respect to the resilience of community water system management but does represent a challenge with respect to measuring the impact of interventions intended to improve water system functionality in selected communities.

f. *Presence of contamination in SWSCs*: While the improvement package improved water quality in intervention communities, detectable microbial contamination remained in a substantive proportion of intervention households and SWSCs. Additional improvement rounds and projects may seek to further control microbial contamination through improvements in source water quality and/or the incorporation of robust water treatment interventions with safe storage containers.

To address implementation challenges associated with organizational and environmental factors (e.g. b, e, f), future efforts may seek to borrow methods and frameworks from implementation science [28]. Specifically, implementation science offers approaches for systematically identifying individual, organizational and environmental factors that can contribute to the successful implementation of improvement packages resulting from iterative CQI processes. Such hybrid approaches, which are increasingly used in the health care field [29], may be instrumental in addressing organizational, logistical, and environmental factors which presented challenges to the application of CQI in the current study context.

Furthermore, ongoing implementation of CQI within and across WaSH implementing organizations may reduce organizational barriers and challenges. Specifically, if such methods become increasingly established in the WaSH sector, many of the relevant skill-sets may be present in implementation organizations at the outset of improvement projects, reducing barriers to start-up and potentially realizing economies of scale.

## Conclusion

This work comprised the first rigorous adaptation of CQI methods to a rural WaSH program, and demonstrated the suitability of this approach for implementing and scaling evidence-based methods for improving the quality and continuity of safe water services in rural northern Ghana. While safe water storage and refresher training are not novel concepts, the adapted

CQI approach allowed the Ghana team to identify and test local adaptations and refinements to the improvement package and validate the effectiveness of these solutions in the local context. These modifications were unlikely to have been identified based on prior knowledge alone, and played an important role in enhancing uptake and performance of the improvement package, as indicated by uptake survey, midline, and endline results.

Based on this initial successful adaptation of CQI to WaSH challenges in a rural lower-middle income country setting, there is emerging evidence to support scaling CQI as a tool for identifying robust and locally appropriate solutions to a broader set of complex WaSH challenges (e.g. consistent and effective drinking water disinfection, sustained sanitation and hygiene uptake) across a broader range of LMIC (and potentially middle- and high-income country) settings in support of progress on SDG 6. There is ample opportunity in such settings to integrate CQI (as an iterative "discovery engine,") with suitable implementation and scale-up frameworks to maximize the impact of successful improvement packages identified through the adapted CQI approach. Furthermore, if these methods become established in the WaSH sector, many implementation challenges can be mitigated and economies of scale realized. Future efforts may focus on building sustainable organizational capacity to develop, implement, monitor, and scale robust, locally appropriate solutions across a broad range of challenges and settings.

## Supporting information

**S1 File. Data collection plan.**
(DOCX)

**S2 File. Detailed description of CQI process steps.**
(DOCX)

**S3 File. CQI Project charter.**
(DOCX)

**S4 File. Survey tools.**
(DOCX)

**S5 File. Selected field protocols and operational definitions.**
(DOCX)

**S6 File. Supporting tables and figures.**
(DOCX)

**S7 File. Structured decision-making tools.**
(DOCX)

**S8 File. Final improvement package.**
(DOCX)

**S9 File. Merged anonymized dataset.**
(XLSX)

## Acknowledgments

The authors gratefully acknowledge The Conrad N. Hilton Foundation and World Vision for support of this work.

## Author Contributions

**Conceptualization:** Michael B. Fisher, Leslie Danquah, Bansaga Saga, Jamie K. Bartram, Rohit Ramaswamy.

**Data curation:** Michael B. Fisher, Leslie Danquah, Zakaria Seidu.

**Formal analysis:** Michael B. Fisher, Allison N. Fechter.

**Funding acquisition:** Jamie K. Bartram, Rohit Ramaswamy.

**Investigation:** Michael B. Fisher, Leslie Danquah, Zakaria Seidu, Allison N. Fechter, Bansaga Saga, Jamie K. Bartram, Kaida M. Liang, Rohit Ramaswamy.

**Methodology:** Michael B. Fisher, Rohit Ramaswamy.

**Project administration:** Michael B. Fisher, Bansaga Saga, Jamie K. Bartram, Kaida M. Liang.

**Resources:** Bansaga Saga.

**Supervision:** Michael B. Fisher, Leslie Danquah, Bansaga Saga, Kaida M. Liang.

**Validation:** Michael B. Fisher, Zakaria Seidu, Allison N. Fechter.

**Visualization:** Michael B. Fisher, Allison N. Fechter, Rohit Ramaswamy.

**Writing – original draft:** Michael B. Fisher, Kaida M. Liang, Rohit Ramaswamy.

**Writing – review & editing:** Michael B. Fisher, Leslie Danquah, Zakaria Seidu, Bansaga Saga, Jamie K. Bartram, Kaida M. Liang, Rohit Ramaswamy.

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
