## [Decision Letter · Decision Letter 0]

19 Aug 2019

PONE-D-19-15425

WaSH CQI: Applying Continuous Quality Improvement methods to Water Service Delivery in four districts of rural northern Ghana

PLOS ONE

Dear Dr. Fisher,

Thank you for submitting your manuscript to PLOS ONE. After careful consideration, we feel that it has merit but does not fully meet PLOS ONE’s publication criteria as it currently stands. Therefore, we invite you to submit a revised version of the manuscript that addresses the points raised during the review process.

Please find detailed comments from the editor as well as reviewers.

We would appreciate receiving your revised manuscript by Oct 03 2019 11:59PM. To enhance the reproducibility of your results, we recommend that if applicable you deposit your laboratory protocols in protocols.io, where a protocol can be assigned its own identifier (DOI) such that it can be cited independently in the future. For instructions see: http://journals.plos.org/plosone/s/submission-guidelines#loc-laboratory-protocols

We look forward to receiving your revised manuscript.

Kind regards,

Michio Murakami

Academic Editor

PLOS ONE

Journal Requirements:

Additional Editor Comments:

This paper includes interesting concept and results; however, some points should be corrected.

In particular, following points should be considered.

1) Please describe all the statistical methods in details in "Methods" section.

The method of logistic regression analysis was described in Results section, but should be moved to Methods.

Furthermore, other statistical test provided in Results (including Supplementary materials) also should be written in Methods.

2) Did the authors perform t-test to compare a proportion of outcome between two groups? In general, t-test is used to compare average values between two groups, while other tests (e.g., chi-square test) are used to test the proportion. Please reconsider statistical methods again.

3) Authors mentioned SDGs in a cover letter, but not in the manuscript. I think the descriptions regarding SDGs written in the cover letter are informative and support the importance of this study. I therefore encourage the authors to include these descriptions and discussions in the manuscript.

4) Please carefully check the author guideline again: https://journals.plos.org/plosone/s/submission-guidelines.

For example, key words are not included in the manuscript. Furthermore, please follow the reference format.

Reviewers' comments:

Reviewer's Responses to Questions

**Comments to the Author**

1. Is the manuscript technically sound, and do the data support the conclusions?

Reviewer #1: Partly

Reviewer #2: Yes

2. Has the statistical analysis been performed appropriately and rigorously? 

Reviewer #1: No

Reviewer #2: Yes

3. Have the authors made all data underlying the findings in their manuscript fully available?

Reviewer #1: No

Reviewer #2: No

4. Is the manuscript presented in an intelligible fashion and written in standard English?

Reviewer #1: No

Reviewer #2: Yes

5. Review Comments to the Author

Reviewer #1: Regarding statistical analysis, authors should show not only the p value but the statistics value such as t-value, z-value.

It is necessary to examine what kind of data should present in the main text. Most of the data was supplied in a supplementary materials and I cannot understand many things just to look the main text.

I cannot imagine the CQI methods concretely. It is better to explain about it.

Authors mentioned the improvement of SWSC. However, there is no data regarding water quality improvement and health issues.

Line 78-79

Authors said that “one-size-fits-all” solutions are rare and does not fit for LIC because the service continuity and water safety depend on context-specific technical, social, geographic, and behavioral factors. However, in Lines 78-79, authors also said that lessons learned from Ghana can be applied to other low- and lower-middle income settings. These are contradictory statements.

Line 148-151

Why the sample size of water sources and that of household survey were different? Why female heads of households were preferred?

Table 2

I cannot understand why the number of communities, water resources of Endline were increased from those of Baseline.

Line 210

Regarding Table S6.1-2, there are some variables with significant difference between control and intervention. These differences may affect to the results. Authors have to explain it. Moreover, Table only showed the p-value. What kind of statistical analysis was done? Authors have to clarify not only p-value also statistical value.

Line 213

Authors said characteristics of intervention and control communities and households were similar at baseline. How can you say so?

Line 218-220

Authors said that many users dipped or scooped despite the presence of tap. How can we know it?

Table 4

What is ROUND 1-4? There was no explanation about it in Method part.

Line 230-231

Why was there a half-year range for midline data collection?

Line 232

What is “105 of 109”.

Line 236-239, 286-288

As the safe water storage containers were delivered only for the intervention households, it is no meaning to compare the proportion of households with safe water storage containers of control group and that of intervention group. Authors have to think the decrease of that ratio within 2 years. Also, I cannot understand the relation of Table 4 and Table 5.

Line 249, 289

There is no definition of high-risk category.

Line 253-255

I cannot understand Table 6.

Line 297-305, 311-313

I cannot find the fact which support these discussions.

Reviewer #2: The paper presents a novel application of Continuous Quality Improvement methods to WASH services in a specific case study in northern Ghana. Despite the local focus of the study and results, it has interest as an applied example of these methodologies to the WASH sector. However, some questions and comments should be considered in a revised version of the paper:

Why the “Survey Tools” are defined for a list of countries and organizations? It has been really applied in this case (Ghana) in that way?

There is a lack of comments about time and resources to perform the proposed overall approach, and specifically regarding data collection (Q243 and Q244 of the first survey, and equivalent ones). I am sure that there are other examples of WASH survey tools less time demanding. Why the authors propose this? A brief review of other options could be useful. In any case, a discussion about scalability of the followed approach is needed.

Reference number 24 should be updated, consider: Requejo et al. “SIASAR: a country-led indicator framework for monitoring the rural water and sanitation sector in Latin America and the Caribbean”, Water Practice and Technology (2017) 12 (2): 372-385.

The DOI numbers of the references should be included.

Finally, it is argued that the data is not available because “they include identifiable personal information including names, GPS coordinates, and other personal information”. I think that this is not a good reason to keep all data supporting the research closed. The authors should anonymize the data sets in order to facilitate the replication of the analysis by the scientific community. Only data used in the analysis is needed. All the extra information should be deleted.

6. PLOS authors have the option to publish the peer review history of their article (what does this mean?). If published, this will include your full peer review and any attached files.

Reviewer #1: No

Reviewer #2: No

---

## [Author Response · Author response to Decision Letter 0]

8 Mar 2020

Response to Reviewers

Dear Editor,

Editor’s and reviewers’ comments have been addressed and the manuscript and supporting information have been updated accordingly. Responses to each comment are listed below and underlined. Where comments could be feasibly addressed as suggested, these changes have been made in the manuscript and supporting information and have been described below. In all but a few cases, comments were addressed as suggested. In those few cases in which comments were not addressed as suggested, an explanation has been provided below. These comments have improved the quality and readability of this manuscript.

If there are any questions about these changes or the responses below, please do not hesitate to contact us.

Best regards,

Michael Fisher.

Michio Murakami

Academic Editor

PLOS ONE

Journal Requirements:

*These changes have been made as suggested. Please let us know if any further changes are needed.*

*These data have been uploaded (Supporting Information File S9).*

*There are no ethical or legal restrictions on sharing a de-identified dataset.*

*The minimal anonymized dataset has been uploaded (Supporting Information File S9).*

*b) The minimal anonymized dataset has been uploaded (Supporting Information File S9).*

Additional Editor Comments:

This paper includes interesting concept and results; however, some points should be corrected.

In particular, following points should be considered.

1) Please describe all the statistical methods in details in "Methods" section.

The method of logistic regression analysis was described in Results section, but should be moved to Methods.

Furthermore, other statistical test provided in Results (including Supplementary materials) also should be written in Methods.

*These changes have been made as suggested (L164-172).*

2) Did the authors perform t-test to compare a proportion of outcome between two groups? In general, t-test is used to compare average values between two groups, while other tests (e.g., chi-square test) are used to test the proportion. Please reconsider statistical methods again.

*These changes have been made as suggested and incorporated in tables and analyses.*

3) Authors mentioned SDGs in a cover letter, but not in the manuscript. I think the descriptions regarding SDGs written in the cover letter are informative and support the importance of this study. I therefore encourage the authors to include these descriptions and discussions in the manuscript.

*These changes have been made as suggested (lines 43-49, 442).*

4) Please carefully check the author guideline again: https://journals.plos.org/plosone/s/submission-guidelines.

For example, key words are not included in the manuscript. Furthermore, please follow the reference format.

*These changes have been made as suggested.*

Reviewers' comments:

Reviewer's Responses to Questions

Comments to the Author

1. Is the manuscript technically sound, and do the data support the conclusions?

Reviewer #1: Partly

Reviewer #2: Yes

2. Has the statistical analysis been performed appropriately and rigorously? 

Reviewer #1: No

Reviewer #2: Yes

*Statistical analyses have been updated as suggested.*

3. Have the authors made all data underlying the findings in their manuscript fully available?

Reviewer #1: No

Reviewer #2: No

*Data have been made fully available.* 

4. Is the manuscript presented in an intelligible fashion and written in standard English?

Reviewer #1: No

Reviewer #2: Yes

*Text has been reviewed and refined throughout to improve readability* 

5. Review Comments to the Author

Reviewer #1: Regarding statistical analysis, authors should show not only the p value but the statistics value such as t-value, z-value.

*Statistical analyses are updated as suggested throughout the manuscript. While it is not always customary to report these additional statistics, they have been added for clarity.*

It is necessary to examine what kind of data should present in the main text. Most of the data was supplied in a supplementary materials and I cannot understand many things just to look the main text.

*Selected tables have been moved back to main text as suggested: specifically S6.13 � Table 2. Moving all supporting information tables to main text would make the manuscript prohibitively lengthy, and thus the remaining tables have been retained in Supporting Information.*

I cannot imagine the CQI methods concretely. It is better to explain about it.

*Additional CQI methods details have been moved back to main text from File S2 as suggested (see lines 105-224)*

Authors mentioned the improvement of SWSC. However, there is no data regarding water quality improvement and health issues.

*The authors note the points regarding water quality, and refer the reader to tables 7a and 7b, which present water quality improvements as the reviewer indicated. While health status could be evaluated between groups based on self-reported diarrhea data, the authors feel that the limited sample size, the indirectness and imprecision of caregiver-reported recall of child diarrhea, and the multiple other factors that may contribute to diarrhea across time and settings (in addition to water quality) make such comparisons less desirable than a straightforward comparison of observed microbial water quality as provided here.*

Line 78-79

Authors said that “one-size-fits-all” solutions are rare and does not fit for LIC because the service continuity and water safety depend on context-specific technical, social, geographic, and behavioral factors. However, in Lines 78-79, authors also said that lessons learned from Ghana can be applied to other low- and lower-middle income settings. These are contradictory statements.

*The text has been clarified to indicate that the CQI process can be adapted to other contexts, but the individual solutions it helps generate do not comprise a one-size-fits-all solution for other contexts per se (Lines 56-74; 80-87).*

Line 148-151

Why the sample size of water sources and that of household survey were different? Why female heads of households were preferred?

*The text has been updated to clarify these points (148-159)*

Table 2

I cannot understand why the number of communities, water resources of Endline were increased from those of Baseline.

*A slightly larger proportion of selected communities were reachable at endline (216) compared to baseline and midline as a result of changes in road conditions and other logistical factors across sampling periods. The text has been updated to clarify this point.*

Line 210

Regarding Table S6.1-2, there are some variables with significant difference between control and intervention. These differences may affect to the results. Authors have to explain it. Moreover, Table only showed the p-value. What kind of statistical analysis was done? Authors have to clarify not only p-value also statistical value.

*The results section has been updated to note the variables for which differences were observed at baseline, and the discussion has been updated to address these points. Tables have been updated to add additional statistical details as suggested.*

Line 213

Authors said characteristics of intervention and control communities and households were similar at baseline. How can you say so?

*Text has been updated to indicate that most variables were similar but that two differences were observed (L245-250).*

Line 218-220

Authors said that many users dipped or scooped despite the presence of tap. How can we know it?

*These practices were observed by enumerators during household visits. The text has been updated to clarify this point (Table 3, L 253-4). *

Table 4

What is ROUND 1-4? There was no explanation about it in Method part.

*This point has been clarified in lines 203-206.*

Line 230-231

Why was there a half-year range for midline data collection?

*Midline data collection was involved (>200 communities) and delayed by some logistical constraints*

Line 232

What is “105 of 109”.

*105 out of the 109 invited WSMTs participated in refresher trainings. This means that of the total 109 WSMTs invited, 105 participated.*

Line 236-239, 286-288

As the safe water storage containers were delivered only for the intervention households, it is no meaning to compare the proportion of households with safe water storage containers of control group and that of intervention group. 

*This is a relatively standard practice to measure implementation fidelity. In other words, as you note we would expect to find safe water storage group in intervention households but not control households. However, control households could have containers meeting the definition of safe storage containers due to implementation errors, “spillover,” or the presence of such containers from some source other than the intervention. Conversely, intervention households might lack safe storage containers if the containers had broken, been lost, or appropriated for other purposes, or had mistakenly been left undelivered or delivered to the wrong households. Finally, errors in appropriately following the correct households and communities over time could have disrupted the expected distribution of containers. Thus, checking to confirm that containers were present in most intervention households and absent in most control households is a useful step to assess implementation fidelity in this study.*

Authors have to think the decrease of that ratio within 2 years. 

*This may be due to loss, breakage, migration, donation, or other factors, as mentioned in the discussion (L342-5).*

Also, I cannot understand the relation of Table 4 and Table 5.

*Table 4 (now Table 5) stratifies by implementation round. Table 5 (now Table 6) stratifies by treatment arm.*

Line 249, 289

There is no definition of high-risk category.

*Text has been updated to specify this category definition: (Line 289-290: E. coli MPN >= 100 CFU/100 mL).*

Line 253-255

I cannot understand Table 6.

*The row captions of this table (Now Table 7) have been edited to specify the distinction between intention to treat (results stratified by assigned study arm) and as-treated or per-protocol (results stratified by observed treatment). Hopefully this makes the table clearer.*

Line 297-305, 311-313

I cannot find the fact which support these discussions.

*The discussion in Lines 332-344 (formerly 297-305) builds on iterative improvements reported in Table 2.

The discussion in lines 361-62 (formerly 311-313) builds upon functionality results reported in Table 8. The discussion text makes this link clear.*

Reviewer #2: The paper presents a novel application of Continuous Quality Improvement methods to WASH services in a specific case study in northern Ghana. Despite the local focus of the study and results, it has interest as an applied example of these methodologies to the WASH sector. However, some questions and comments should be considered in a revised version of the paper:

Why the “Survey Tools” are defined for a list of countries and organizations? It has been really applied in this case (Ghana) in that way?

*These CQI survey tools were adapted from monitoring and evaluation tools deployed in multiple settings. This is why multiple country options are listed in each survey. Furthermore, water sources implemented by a variety of different entities and organizations were encountered. This is why multiple implementer options are listed in each survey.A note has been added to Supporting Information File S4 to clarify this point.*

There is a lack of comments about time and resources to perform the proposed overall approach, and specifically regarding data collection (Q243 and Q244 of the first survey, and equivalent ones). I am sure that there are other examples of WASH survey tools less time demanding. Why the authors propose this? A brief review of other options could be useful. In any case, a discussion about scalability of the followed approach is needed.

*Median survey completion times were approximately 10, 20, and 25 minutes for waterpoint, WSMT, and household surveys, respectively. These details have been added to the methods section. Cost of monitoring is not discussed in this work, but was approximately $100 per community, and was largely the cost of fuel and personnel effort. Given the long distances often needed to travel to communities, the incremental cost of each additional minute of data collection was small relative to the cost of traveling to communities. Thus, the decision to use more detailed surveys to capture important process and outcome variables at baseline, midline, and endline was made, vs the decision to use more streamlined tools. furthermore, detailed data collection on process and outcome variables was of particular interest given the lack of high-quality evidence on WaSH CQI in rural LMIC settings. These points have also been added to the discussion (L 385-396).*

Reference number 24 should be updated, consider: Requejo et al. “SIASAR: a country-led indicator framework for monitoring the rural water and sanitation sector in Latin America and the Caribbean”, Water Practice and Technology (2017) 12 (2): 372-385.

*This reference has been added as suggested. The original reference has also been retained*

The DOI numbers of the references should be included.

*These are included in the citation manager, where available, but are not printed in the current citation export format. PLoS ONE uses Vancouver format, which does not include DOI numbers.*

Finally, it is argued that the data is not available because “they include identifiable personal information including names, GPS coordinates, and other personal information”. I think that this is not a good reason to keep all data supporting the research closed. The authors should anonymize the data sets in order to facilitate the replication of the analysis by the scientific community. Only data used in the analysis is needed. All the extra information should be deleted.

*This has been done as suggested (Supporting Information File S9).*

6. PLOS authors have the option to publish the peer review history of their article (what does this mean?). If published, this will include your full peer review and any attached files.

Do you want your identity to be public for this peer review? For information about this choice, including consent withdrawal, please see our Privacy Policy.

Reviewer #1: No

Reviewer #2: No

---

## [Decision Letter · Decision Letter 1]

10 Apr 2020

PONE-D-19-15425R1

WaSH CQI: Applying Continuous Quality Improvement methods to Water Service Delivery in four districts of rural northern Ghana

PLOS ONE

Dear Dr. Fisher,

Thank you for submitting your manuscript to PLOS ONE. After careful consideration, we feel that it has merit but does not fully meet PLOS ONE’s publication criteria as it currently stands. Therefore, we invite you to submit a revised version of the manuscript that addresses the points raised during the review process.

We would appreciate receiving your revised manuscript by May 25 2020 11:59PM. To enhance the reproducibility of your results, we recommend that if applicable you deposit your laboratory protocols in protocols.io, where a protocol can be assigned its own identifier (DOI) such that it can be cited independently in the future. For instructions see: http://journals.plos.org/plosone/s/submission-guidelines#loc-laboratory-protocols

We look forward to receiving your revised manuscript.

Kind regards,

Michio Murakami

Academic Editor

PLOS ONE

Additional Editor Comments (if provided):

Some numbers in main text should be corrected following the changes of tables.

L280 (in the file with marked changes)

84%=>86%

L282

52%=>57%

L347

26%=>30%

L348

42%=>43%

Please carefully check all the numbers again.

Reviewers' comments:

Reviewer's Responses to Questions

**Comments to the Author**

1. If the authors have adequately addressed your comments raised in a previous round of review and you feel that this manuscript is now acceptable for publication, you may indicate that here to bypass the “Comments to the Author” section, enter your conflict of interest statement in the “Confidential to Editor” section, and submit your "Accept" recommendation.

Reviewer #1: All comments have been addressed

Reviewer #2: All comments have been addressed

2. Is the manuscript technically sound, and do the data support the conclusions?

Reviewer #1: Yes

Reviewer #2: (No Response)

3. Has the statistical analysis been performed appropriately and rigorously? 

Reviewer #1: Yes

Reviewer #2: (No Response)

4. Have the authors made all data underlying the findings in their manuscript fully available?

Reviewer #1: Yes

Reviewer #2: (No Response)

5. Is the manuscript presented in an intelligible fashion and written in standard English?

Reviewer #1: Yes

Reviewer #2: (No Response)

6. Review Comments to the Author

Reviewer #1: I confirmed all the revisions and agreed them.

It is better to use italic character, when showing statistics　(ex. p-value).

Reviewer #2: The new versión is much more clear. Most of the comments has been properly considered.

One minor comment. I propose to reduce the number of decimals in Table 2 from 6 to 3. They are estimations and confidence intervals, not precis mechanical measurements. The table footnote with two stars could be removed (in this table).

7. PLOS authors have the option to publish the peer review history of their article (what does this mean?). If published, this will include your full peer review and any attached files.

Reviewer #1: No

Reviewer #2: No

---

## [Author Response · Author response to Decision Letter 1]

7 May 2020

Response to Reviewers Updated 5.7.2020

Dear Editor,

Editor’s and reviewers’ additional comments have been addressed and the manuscript and supporting information have been updated accordingly. Responses to each comment are listed below and marked with double asterisks (**) before and after. These comments have improved the quality and readability of this manuscript.

If there are any questions about these changes or the responses below, please do not hesitate to contact us.

Best regards,

Michael Fisher.

PONE-D-19-15425R1

WaSH CQI: Applying Continuous Quality Improvement methods to Water Service Delivery in four districts of rural northern Ghana

PLOS ONE

Additional Editor Comments (if provided):

Some numbers in main text should be corrected following the changes of tables.

L280 (in the file with marked changes)

84%=>86%

L282

52%=>57%

L347

26%=>30%

L348

42%=>43%

**These changes have been made as suggested**

Please carefully check all the numbers again.

**OK**

Reviewers' comments:

Reviewer's Responses to Questions

Comments to the Author

1. If the authors have adequately addressed your comments raised in a previous round of review and you feel that this manuscript is now acceptable for publication, you may indicate that here to bypass the “Comments to the Author” section, enter your conflict of interest statement in the “Confidential to Editor” section, and submit your "Accept" recommendation.

Reviewer #1: All comments have been addressed

Reviewer #2: All comments have been addressed

2. Is the manuscript technically sound, and do the data support the conclusions?

Reviewer #1: Yes

Reviewer #2: (No Response)

3. Has the statistical analysis been performed appropriately and rigorously? 

Reviewer #1: Yes

Reviewer #2: (No Response)

4. Have the authors made all data underlying the findings in their manuscript fully available?

Reviewer #1: Yes

Reviewer #2: (No Response)

5. Is the manuscript presented in an intelligible fashion and written in standard English?

Reviewer #1: Yes

Reviewer #2: (No Response)

6. Review Comments to the Author

Reviewer #1: I confirmed all the revisions and agreed them.

It is better to use italic character, when showing statistics　(ex. p-value).

**These changes have been made: p-values have been italicized.**

Reviewer #2: The new versión is much more clear. Most of the comments has been properly considered.

One minor comment. I propose to reduce the number of decimals in Table 2 from 6 to 3. They are estimations and confidence intervals, not precis mechanical measurements. The table footnote with two stars could be removed (in this table). 

**This change has been made. Significant figures have been reduced as suggested.** 

7. PLOS authors have the option to publish the peer review history of their article (what does this mean?). If published, this will include your full peer review and any attached files.

Do you want your identity to be public for this peer review? For information about this choice, including consent withdrawal, please see our Privacy Policy.

Reviewer #1: No

Reviewer #2: No

---

## [Editor Report · Decision Letter 2]

12 May 2020

WaSH CQI: Applying Continuous Quality Improvement methods to Water Service Delivery in four districts of rural northern Ghana

PONE-D-19-15425R2

Dear Dr. Fisher,

We are pleased to inform you that your manuscript has been judged scientifically suitable for publication and will be formally accepted for publication once it complies with all outstanding technical requirements.

With kind regards,

Michio Murakami

Academic Editor

PLOS ONE
---

## [Editor Report · Acceptance letter]

12 Jun 2020

PONE-D-19-15425R2 

WaSH CQI: Applying Continuous Quality Improvement methods to Water Service Delivery in four districts of rural northern Ghana 

Dear Dr. Fisher:

I'm pleased to inform you that your manuscript has been deemed suitable for publication in PLOS ONE. Congratulations! Your manuscript is now with our production department. 

Kind regards, 

on behalf of

Dr. Michio Murakami 

Academic Editor

PLOS ONE